# Improving outcomes for neonates with gastroschisis in low-income and middle-income countries: a systematic review protocol

Naomi J Wright,[1] Monica Langer,[2] Irena CF Norman,[3] Melika Akhbari,[3] Q Eileen Wafford,[4] Niyi Ade-Ajayi,[5] Justine Davies,[1] Dan Poenaru,[6] Nick Sevdalis,[7] Andy Leather[1]

► Additional material is published online only. To view please visit the journal online (http://dx.doi.org/10.1136/bmjpo-2018-000392).

For numbered affiliations see end of article.

**Correspondence to**
Naomi J Wright; naomiwright@doctors.org.uk

## ABSTRACT

**Introduction** There is a significant disparity in outcomes for neonates with gastroschisis in high-income countries (HICs) compared with low-income and middle-income countries (LMICs). Many LMICs report mortality rates between 75% and 100% compared with <4% in HICs.

**Aim** To undertake a systematic review identifying postnatal interventions associated with improved outcomes for gastroschisis in LMICs.

**Methods and analysis** Three search strings will be combined: (1) neonates; (2) gastroschisis and other gastrointestinal congenital anomalies requiring similar surgical care; (3) LMICs. Databases to be searched include MEDLINE, EMBASE, Scopus, Web of Science, ProQuest Dissertations and Thesis Global, and the Cochrane Library. Grey literature will be identified through Open-Grey, ClinicalTrials.gov, WHO International Clinical Trials Registry and ISRCTN registry (Springer Nature). Additional studies will be sought from reference lists of included studies. Study screening, selection, data extraction and assessment of methodological quality will be undertaken by two reviewers independently and team consensus sought on discrepancies. The primary outcome of interest is mortality. Secondary outcomes include complications, requirement for ventilation, parenteral nutrition duration and length of hospital stay. Tertiary outcomes include service delivery and implementation outcomes. The methodology of the studies will be appraised. Descriptive statistics and outcomes will be summarised and discussed.

**Ethics and dissemination** Ethical approval is not required since no new data are being collected. Dissemination will be via open access publication in a peer-reviewed medical journal and distribution among global health, global surgery and children's surgical collaborations and international conferences.

**Conclusion** This study will systematically review literature focused on postnatal interventions to improve outcomes from gastroschisis in LMICs. Findings can be used to help inform quality improvement projects in low-resource settings for patients with gastroschisis. In the first instance, results will be used to inform a Wellcome Trust-funded multicentre clinical interventional study aimed at improving outcomes for gastroschisis across sub-Saharan Africa.

**PROSPERO registration number** CRD42018095349.

## INTRODUCTION

Congenital anomalies are estimated to be the fifth leading cause of death in under 5 year olds globally.[1] Gastroschisis (a condition where the intestines protrude through a hole in the abdominal wall at birth) is one of the the most common congenital anomalies and has been increasing in incidence globally.[2–6] It occurs in approximately 1 in 2000 births. With an estimated 32 million births per year in sub-Saharan Africa (SSA), we would expect 16 000 neonates with gastroschisis to be born in the region annually. Indeed, Paediatric Surgeons across SSA report receiving between 1 and 15 cases per month.[7] Since the 1960s, mortality from gastroschisis has fallen in high-income countries (HICs) to <5% today.[7] Mortality has fallen to a lesser extent in middle-income countries. Recent literature reports mortality rates of 80%, Iran; 36%, Turkey; and 6%–8%, Thailand.[8–11] In low-income countries, mortality remains high, with many SSA countries reporting mortality rates of 75%–100%.[12–15]

Management of gastroschisis varies widely. The most common interventions in HICs are primary closure in the operating room or use of a preformed silo with gradual intestinal reduction and delayed closure, often at the cotside without general anaesthetic.[16] Systematic reviews report comparable outcomes for both methods in HICs, but with lower ventilation requirements associated with the use of a preformed silo.[17 18] Preformed silo use has additional benefits for LMICs: it is low-technology; avoids neonatal anaesthesia and surgery; can be applied at the cotside by any trained healthcare personnel and reduces intensive care requirements due to lower

intra-abdominal pressures.[17] This is advantageous in low-resource settings where there is variable availability of paediatric surgeons, deficient intensive care facilities, and safety of neonatal anaesthesia and surgery is limited by the lack of trained staff and resources.[19 20]

However, preformed silos are expensive and have been largely unavailable in LMICs and hence alternative strategies have been devised.[11] Examples include use of an Alexis Wound Retractor as an alternative to the preformed silo, primary reduction at the cotside (Bianchi technique) and umbilical turban and flap closure.[21–28] Furthermore, antenatal diagnosis, delivery in a tertiary paediatric surgery centre, prehospital management, neonatal resuscitation and nutrition are all fundamental components of care that impact survival.[15] Interventions aimed at improving one or more of these components has the potential to significantly improve outcomes.

Some centres within low-resource settings have managed to achieve better survival from gastroschisis and other similar congenital anomalies involving the gastrointestinal tract using one or more of the above interventions. However, to our knowledge, there has never been a systematic review to collate and analyse such evidence from LMIC settings. Hence, the focus of this systematic review is to identify postnatal prehospital and inhospital interventions aimed at improving outcomes for neonates with gastroschisis in LMICs. This information is vital to inform quality improvement projects aimed at improving survival from gastroschisis in LMICs. In the first instance, the results of this review will be used in the design of a Wellcome Trust-funded multicentre clinical interventional study aimed at reducing mortality from gastroschisis in seven tertiary paediatric surgery centres in SSA.

In this review, an 'intervention' is defined as any action taken to improve a patient's medical condition. This includes specific interventions for gastroschisis and generic interventions used for a wider range of congenital anomalies involving the gastrointestinal tract, which may also be beneficial for patients with gastroschisis. The review will not include antenatal interventions since another systematic review is currently in progress focused on this topic.

## METHODS AND ANALYSIS

Preferred Reporting Items for Systematic Review and Meta-Analysis Protocols (PRISMA-P) 2015 guidelines have been followed in this protocol.[29] Online supplementary file 1 details the PRISMA-P checklist and where each of the items is addressed in this protocol. If amendments to the protocol occur, they will be reported in the publication of the results.

### Aim

To identify postnatal prehospital and inhospital interventions associated with improved outcomes for neonates with gastroschisis in LMICs.

### Objectives

1. To identify studies that evaluate postnatal interventions to improve mortality and morbidity for neonates with gastroschisis in LMICs.
2. To identify generic surgical care interventions used in LMICs to manage neonates with a wider range of structural congenital anomalies involving the gastrointestinal tract, which may be transferable to the care of neonates with gastroschisis.
3. To critically appraise the methodological quality of the evidence.
4. To provide an evidence-based summary of the condition-specific and generic neonatal surgical care interventions associated with improved outcomes for gastroschisis in LMICs to inform clinical practice and future studies.

### Search strategy

A medical research librarian developed the search strategy in collaboration with members of the review team. The search was optimised by testing the sensitivity and specificity of the search terms during the development phase and revising the search strategy accordingly. The search strategy consists of controlled vocabulary and keywords for (1) the population—neonates, (2) the conditions—gastroschisis and a selection of structural congenital anomalies involving the gastrointestinal tract requiring a similar package of neonatal surgical care and (3) the context—LMICs.

Neonates are defined as infants within the first 28 days of life. Terms for structural congenital anomalies involving the gastrointestinal tract are outlined in search string 2 (table 1). These were derived from consensus among the authors according to what conditions may use similar neonatal surgical care as gastroschisis and thus have relevant transferable interventions. The third search string includes all countries listed as low-income or middle-income by the World Bank in 2018 and the varying terminology utilised to describe LMICs.[30] Individual countries and major cities will be included.

The search strategy was developed in MEDLINE (online supplementary document 2). A highly sensitive search will be undertaken by employing truncation and wildcards and applying the Unqualified Searches (MP) tag to search text words. The search strategy will be adapted to MEDLINE (Ovid), EMBASE (Elsevier), Scopus (Elsevier), Web of Science (Clarivate Analytics), ProQuest Dissertations & Thesis Global, and the Cochrane Library. Each database will be searched from the date of inception. The search will not be restricted based on language or study design. Only human studies will be included. Literature reviews and reference lists of included studies will be searched for further studies suitable for inclusion.

Grey literature will be included to help mitigate the risks of publication bias and to identify the latest progress in the field. We will identify unpublished studies by searching the following grey literature sources: Open-Grey, ClinicalTrials.gov, WHO International Clinical

**Table 1** Three search strings to be utilised to identify studies to be included in the systematic review

| Search string 1 | Search string 2 | Search string 3 |
|---|---|---|
| Newborn neonate | Congenital anomalies, congenital abnormalities, congenital malformation, birth defects, gastroschisis, exomphalos, omphalocele, abdominal wall defect, intestinal atresia, apple peel syndrome, duodenal atresia, duodenal obstruction, duodenal web, jejunal atresia, jejuno-ileal atresia, ileal atresia, colonic atresia, anorectal malformation, anorectal stenosis, imperforate anus, anal atresia, malrotation, volvulus, congenital diaphragmatic hernia, oesophageal atresia, tracheo-oesophageal fistula, Hirschsprung's disease, aganglionosis | Low income countries, middle income countries, LMICs, LAMI, LMI, low resource settings, resource limited setting, less resourced communities, developing countries, underdeveloped countries, third world countries, developing nations, low income nation, sub-Saharan Africa, Afghanistan, Albania, Algeria, American Samoa, Angola, Argentina, Armenia, Azerbaijan, Bangladesh, Belarus, Belize, Benin, Bhutan, Bolivia, Bosnia and Herzegovina, Botswana, Brazil, Bulgaria, Burkina Faso, Burundi, Cabo Verde, Cambodia, Cameroon, Central African Republic, Chad, China, Columbia, Comoros, Democratic Republic of the Congo, DRC, Republic of the Congo, Costa Rica, Cote d'Ivoire, Ivory Coast, Croatia, Cuba, Djibouti, Dominica, Ecuador, Egypt, El Salvador, Equatorial Guinea, Eritrea, Ethiopia, Fiji, Gabon, Gambia, Georgia, Ghana, Grenada, Guatemala, Guinea, Guinea-Bissau, Guyana, Haiti, Honduras, India, Indonesia, Islamic Republic of Iran, Iraq, Jamaica, Jordan, Kazakhstan, Kenya, Kiribati, Democratic People's Republic of Korea, Kosovo, Kyrgyz Republic, Lao PDR, Laos, Lebanon, Lesotho, Liberia, Libya, Macedonia Republic, Madagascar, Malawi, Malaysia, Maldives, Mali, Marshall Islands, Mauritania, Mauritius, Mexico, Micronesia, Moldova, Mongolia, Montenegro, Morocco, Mozambique, Myanmar, Namibia, Nauru, Nepal, Nicaragua, Niger, Nigeria, Pakistan, Panama, Papua New Guinea, Paraguay, Peru, Philippines, Romania, Russian Federation, Rwanda, Samoa, Sao Tome and Principe, Senegal, Serbia, Sierra Leone, Solomon Islands, Somalia, Somaliland, South Africa, Sri Lanka, St. Lucia, Saint Lucia, St. Vincent and Grenadines, Saint Vincent and the Grenadines, Sudan, Suriname, Swaziland, Syrian Arab Republic, Syria, Tajikistan, Tanzania, Thailand, Timor-Leste, East Timor, Togo, Tonga, Tunisia, Turkey, Turkmenistan, Tuvalu, Uganda, Ukraine, Uzbekistan, Vanuatu, Venezuela, Vietnam, West Bank and Gaza, Republic of Yemen, Zambia, Zimbabwe. Kabul, Porto-Novo, Hogbonou, Adjace, Cotonou, Kutonu, Ouagadougou, Ouaga, Bujumbura, Usumbura, Phnom Penh, Bangui, Bangi, N'Djamena, Ndjamena, Fort Lamy, Moroni, Kinshasa, Asmara, Asmera, Addis Ababa, Addis Abeba, Banjul, Bathurst, Conakry, Bissau, Port-au-Prince, Monrovia, Antananarivo, Tananarive, Tana, Lilongwe, Bamako, Maputo, Lourenco Marques, Kathmandu, Niamey, Kigali, Freetown, Free-town, Mogadishu, Xamar, Hamar, Muqdisho, Maqadishu, Juba, Dodoma, Dar es Salaam, Lome, Kampala, Harare, Salisbury, Yerevan, Dhaka, Dacca, Thimphu, Thimbu, Sucre, Charcas, La Plata, Chuquisaca, La Paz, Praia, Yaounde, Jaunde, Brazzaville, Yamoussoukro, Cairo, Accra, Tegucigalpa, Tegus, New Delhi, Jakarta, Nairobi, South Tarawa, Tarawa Teinainano, Pristina, Prishtina, Bishkek, Pishpek, Frunze, Vientiane, Maseru, Nouakchott, Palikir, Chisinau, Kishinev, Rabat, Nay Pyi Taw, Naypyidaw, Nepranytau, Naypyitaw, Kyetpyay, Pyinmana, Kyatpyay, Pyinmana, Yangon, Rangoon, Managua, Abuja, Lagos, Islamabad, Port Moresby, Moresby, Pom Town, Manila, Apia, Dakar, Honiara, Jayawardenepura, Jayewardenepura, Khartoum, Mbabane, Embabane, Lobamba, Damascus, Dushanbe, Dyushambe, Stalinabad, Dili, Kyiv, Kiev, Tashkent, Toshkent, Port Vila, Hanoi, Ha Noi, Sana'a, Sanaa, Sana, Lusaka, Ulaanbaatar, Ulan-Bator, Luanda, Tbilisi, Amman |

Trials Registry and ISRCTN registry (Springer Nature). These were selected since they are major sources of grey literature for the biosciences providing comprehensive coverage. The following publication types will be included: dissertations/theses, books/book chapters, conference abstracts, editorials/letters/ comments, newspapers/trade journals, literature reviews and research in progress. Experts in the field will be contacted to identify any ongoing research on this subject, which has yet to be published. The authors of identified grey literature will be contacted for a full report of data and findings where available.

### Inclusion/exclusion criteria
#### Conditions
Conditions to be included are listed in column 2 of table 1 (search string 2). Neonates with these structural congenital anomalies involving the gastrointestinal tract commonly present with a life-threatening emergency requiring a similar package of care within the neonatal period.

Structural congenital anomalies involving the gastrointestinal tract to be excluded from the search criteria include biliary atresia, choledochal cyst and all other conditions not listed under search string 2. These conditions often present outside of the neonatal period.

#### Setting and participants
Studies containing preterm and term neonates presenting within the first 28 days of life with gastroschisis or one of the structural congenital anomalies listed in search string 2 will be included. Studies including just patients who have previously received care and re-presented with a complication or need for further intervention will be excluded. Only studies which have been undertaken in LMICs will be included.

#### Interventions
All prehospital and inhospital postnatal interventions for the care of neonates with gastroschisis in LMICs will be included. Generic interventions related to the care of neonates with a structural congenital anomaly involving

| Table 2 | Clavien-Dindo classification of complications[31] |
|---|---|
| Grade | Definition |
| I | Any deviation from the normal postoperative course without the need for pharmacological treatment or surgical, endoscopic and radiological interventions. Allowed therapeutic regimes are as follows: drugs as antiemetics, antipyretics, analgesics, diuretics, electrolytes, parenteral nutrition and physiotherapy. This grade also includes wound infections opened at the bedside |
| II | Requiring pharmacological treatment with drugs other than such allowed for grade I complications. Blood transfusions are also included |
| III | Requiring surgical, endoscopic or radiological intervention |
| IIIa | Intervention not under general anaesthesia |
| IIIb | Intervention under general anaesthesia |
| IV | Life-threatening complication requiring ICU management |
| IVa | Single organ dysfunction (including dialysis) |
| IVb | Multiorgan dysfunction |
| V | Death of a patient |

| Table 3 | Definition of implementation outcomes[40] |
|---|---|
| Implementation outcome | Definition |
| Acceptability | Perception among stakeholders that the new intervention is agreeable |
| Adoption | Intention to apply new intervention |
| Appropriateness | Perceived relevance of the intervention for the setting and problem |
| Feasibility | Extent to which an intervention can be applied |
| Fidelity | The proportion of management protocol components completed as intended |
| Coverage | The proportion of eligible patients who actually receive the intervention |
| Cost | Costs of the intervention, including the delivery strategy |
| Sustainability | Extent to which a new intervention becomes routinely available/is maintained postintroduction |

the gastrointestinal tract will be included. Antenatal interventions will be excluded because they are currently being evaluated in a separate systematic review.

Types of interventions will be categorised into specific interventions for neonates with gastroschisis and generic neonatal surgical care interventions for structural congenital anomalies involving the gastrointestinal tract. Generic interventions will be subcategorised into the following: prehospital care and transportation, place of delivery, neonatal resuscitation and care, staffing, access to parenteral nutrition and other. Operative interventions related specifically and solely to a condition other than gastroschisis will be excluded. For example, operative techniques for oesophageal atresia or anorectal malformation.

### Outcome measures
The primary outcome of the review will be mortality. This will include all-cause inhospital mortality, mortality within the neonatal period (within 28 days of life) and 30-day postintervention mortality. Secondary outcomes will include: complications (post-primary intervention (primary intervention is defined as the first intervention the neonate received for bowel coverage, including the application of a preformed silo at the cotside.)), requirement for ventilation (yes/no, number of days), duration of parenteral nutrition (days) and length of hospital stay (days). Complications will be determined using the Clavien-Dindo classification (table 2).[31]

An improved primary or secondary outcome will be defined as a significant difference with a p value <0.05.

Tertiary outcomes include service delivery and implementation outcomes (table 3). Implementation strategies will also be analysed. An implementation strategy is defined as the method(s) or technique(s) used to enhance the adoption, implementation and sustainability of a clinical programme or practice.[32]

### Study screening
References identified through the electronic search engines will be entered into Covidence and duplicates removed.[33] Two reviewers will independently screen the titles and abstracts of all references. All potentially relevant articles will have the full text reviewed in duplicate against the eligibility criteria. This will include the full text of articles in languages other than English, which will be translated. Inter-rater reliability will be assessed following the screening of the first 50 abstracts through a review of the decisions made and discussion among the wider authorship. Consensus will be sought on the hierarchy of reasons for rejecting studies to ensure consistency among the study team. Any discrepancies during the screening process will be highlighted in Covidence and will be resolved by consensus with the wider authorship group. All reviewers are trained in systematic review methods. The search results will be represented using a PRISMA flowchart.[34]

### Data extraction
Data will be extracted in duplicate by two reviewers and entered into a predetermined data collection form. Data will be collected on the study type, country, year of publication, journal of publication, authors' names, number of patients, patient demographics (including proportion with simple and complex gastroschisis (patients with bowel necrosis, perforation, atresia or closing/closed gastroschisis)), gestational age, weight, time from birth to presentation at the study hospital and American Association of Anesthesiologists

Score at the time of primary intervention), prehospital and inhospital intervention(s), implementation strategy where relevant, primary and secondary clinical outcomes as detailed above, service delivery outcomes and implementation outcomes if available. The two data extraction databases will be compared and any discrepancies discussed with the wider authorship to determine consensus. The data collection form will be sent to investigators of unpublished studies to obtain such data where available.

## Data synthesis

Descriptive statistics will be used to present the interventions and outcomes in results tables, accompanied by a narrative synthesis. Interventions will be categorised into gastroschisis-specific and generic neonatal surgical care, with the latter being further subcategorised as detailed above. Because a wide range of interventions (often a group of interventions combined) and outcomes will be evaluated, it is unlikely that a meta-analysis will be feasible. However, if there is appropriate data, a meta-analysis will be undertaken. Appropriate data will be defined as two or more studies comparing the mortality between two or more of the same interventions so we can pool the data and perform a meta-analysis. For example, two or more studies comparing the mortality outcome between intervention 'a' with intervention 'b'. Meta-analysis will be undertaken in Stata and results presented using a forest plot. If there are over 10 studies in the meta-analysis, a funnel plot will be undertaken to assess publication bias and a Galbraith plot to investigate heterogeneity in effect sizes. The quality of evidence will be assessed following GRADE guidelines.[35]

## Methodological quality appraisal and bias assessment

The methodological quality of the individual studies will be assessed and the findings summarised in a table to aid interpretation. This will be incorporated into the narrative synthesis. Cochrane Risk of Bias for Non-Randomised Studies of Interventions and the revised tool for Risk of Bias in randomised trials (RoB 2.0) will be used to assess quantitative studies.[36 37] This will be undertaken by two reviewers independently and team consensus sought for discrepancies.

## Patient and public involvement

Patient and Public Involvement (PPI) was discussed among the team. On this occasion it was decided to be unfeasible since the relevant PPI members are most commonly parents of neonates with gastroschisis living in LMICs. However, we acknowledge the importance of PPI and will continue to seek ways to involve patients, parents and public in this ongoing gastroschisis research.

## DISCUSSION

To our knowledge, this will be the first systematic review focused on postnatal interventions to improve outcomes from gastroschisis in LMICs. Such a review is vital to address the current outcome disparities, with many neonates with gastroschisis dying in LMICs and the majority surviving in HICs.[11 15] It is hoped that lessons learnt in centres with better outcomes within LMICs can be evaluated and shared among the global community to improve outcomes and inform future interventional studies. A wider range of congenital anomalies involving the gastrointestinal tract will be incorporated into the study to help identify generic neonatal surgical care interventions that have the potential to also improve outcomes for neonates with gastroschisis. This information may also help to inform clinical practice for a wider range of structural congenital anomalies involving the gastrointestinal tract. The systematic review may also highlight areas for improvement in HICs, such as cost reduction.

## Strengths and limitations

This study is unique in its focus and comprehensive search strategy incorporating both original articles, grey literature, published and unpublished work. Incorporating a wider range of structural congenital anomalies within the search strategy will help to identify neonatal surgical care interventions used in LMICs that could be beneficial for patients with gastroschisis. Identified articles in languages other than English will be translated so they can be included within the review.

Although the search strategy has been designed to be optimally inclusive, it is possible that articles could be missed. The initial search will be undertaken in English and some articles without an English translation of the title or abstract could be missed. This systematic review will only include studies undertaken within LMICs; it may be possible that low-technology interventions used within HICs could also benefit gastroschisis care in low-resource settings.

## Ethics and dissemination

This systematic review will analyse previously published historical data, thus does not require ethical approval.

The results will be submitted for open access publication in a peer-reviewed medical journal. The publication will be disseminated among the PaedSurg Africa Research Collaboration and Global PaedSurg Research Collaboration consisting of hundreds of children's surgical care providers across the globe.[38] It will also be shared among members of the Global Initiative for Children's Surgery, which includes all members of the multidisciplinary team caring for neonates with gastroschisis and international organisations, policy-makers and representatives from the WHO.[39] Results will be disseminated using social media. Findings will be presented internationally with a focus on global health, global surgery, paediatric and paediatric surgical conferences in LMICs.

The results will help to inform the development of an interventional care bundle to be evaluated in a Wellcome Trust-funded multicentre interventional study aimed at improving survival in neonates with

gastroschisis in LMICs. This will be undertaken in seven tertiary paediatric surgery centres across SSA between 2018 and 2020.

**Author affiliations**
[1]King's Centre for Global Health and Health Partnerships, School of Population Health and Environmental Sciences, King's College London, London, UK
[2]Northwestern University, Chicago, USA
[3]GKT School of Medical Education, King's College London, London, UK
[4]Galter Health Sciences Library and Learning Center, Northwestern University, Chicago, Illinois, USA
[5]Paediatric Surgery Department, King's College Hospital, London, UK
[6]McGill University, Montreal, Canada
[7]Centre for Implementation Science, King's College London, London, UK

**Acknowledgements** The authors thank the library services at King's College London, UK; Northwestern University, Chicago, USA; and Elena Guadagno, MLIS, and Alex Amar, MLIS, at McGill University Health Centre (MUHC), Montreal, Canada for their help with the literature search.

**Contributors** NJW and ML: conceived the idea for the study and designed the protocol. NJW: drafted the protocol, designed the search strategy in conjunction with QEW and is the guarantor of the review. QEW: contributed to the design of the search strategy and undertook the search. All authors: provided input on the study design and protocol development and contributed to the final manuscript.

**Funding** NW's research is funded by the Wellcome Trust as part of a Clinical PhD in Global Health at King's Centre for Global Health and Health Partnerships, London, UK. NS' research is supported by the National Institute for Health Research (NIHR) Collaboration for Leadership in Applied Health Research and Care South London at King's College Hospital NHS Foundation Trust. NS is also a member of King's Improvement Science, which is part of the NIHR CLAHRC South London and comprises a specialist team of improvement scientists and senior researchers based at King's College London. Its work is funded by King's Health Partners (Guy's and St Thomas' NHS Foundation Trust, King's College Hospital NHS Foundation Trust, King's College London and South London and Maudsley NHS Foundation Trust), Guy's and St Thomas' Charity, the Maudsley Charity and the Health Foundation.

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
