## [Reviewer comments · BMJ Paediatrics Open]

ARTICLE DETAILS

TITLE (PROVISIONAL)	Improving outcomes for neonates with gastroschisis in low- and middle-income countries: a systematic review protocol
AUTHORS	

VERSION 1 – REVIEW

REVIEWER	Reviewer name: Aminde, Leopold Ndemnge Institution and Country: The University of Queensland, School of Public Health, Australia Competing interests: None
REVIEW RETURNED	30-Oct-2018

GENERAL COMMENTS	Wright and colleagues present a proposal for a systematic review to assess post-natal interventions geared towards improving outcomes in neonates with gastroschisis and related GI anomalies in LMICs. The review team must be commended for exploring such an important area as LMICs especially Africa, carry the greatest burden of neonatal and infant mortality. The paper is mostly well written, and has been registered in PROSPERO. Just a some minor comments; 1) Background; This presents a good overview of the problem of comparatively high rates of death in LMIC compared to HIC for gastrochisis. Could authors include a paragraph or so more clearly linking this burden up with the justification and need for the review? This [link] doesn't seem to come out very clearly. 2) Methods; Clearly described. - In page 4 line 21, could authors include 'Protocols' so that the statement reads "Preferred Reporting Items for Systematic review and Meta-Analysis Protocols"(PRISMA-P) ?- In the data synthesis, given the anticipated variation in interventions authors are likely to meet, you mention descriptive statistics would be reported. This should be accompanied with a narrative synthesis. Secondly, authors mention a meta-analysis is not planned, but in case there is 'appropriate data' they will progress with a meta-analysis. Could you be more specific and or elaborate on what 'appropriate data' you will be interested in so as to perform a meta-analysis? - Consider discussing your process of data management.
---

REVIEWER	Reviewer name: Tamara Fitzgerald Institution and Country: Duke University, USA Competing interests: I have no competing interests or financial disclosures.
REVIEW RETURNED	07-Nov-2018

GENERAL COMMENTS	The authors present a strategy for performing a systematic review of gastroschisis management and outcomes in low- and middle-
--

	income countries. The review is an important task to undertake and their protocol appears well thought out.
--	---

REVIEWER	Reviewer name: Nancy Butcher, PhD Institution and Country: Senior Research Associate, Child Health Evaluative Sciences, The Hospital for Sick Children, Peter Gilgan Centre for Research & Learning, Toronto, Ontario, Canada. Competing interests: None to declare.
REVIEW RETURNED	07-Nov-2018

GENERAL COMMENTS	Neonates with gastroschisis in LMICs face poorer outcomes than those in high-income countries. The authors describe a protocol for a systematic review to identify postnatal interventions for neonates with gastroschisis in LMICs that ultimately may help inform clinical strategies for this population. The protocol would benefit from providing some additional information and clarifications, as follows: GENERAL  -The authors indicate the protocol was prepared using PRISMA-P checklist. The authors should provide the completed PRISMA-P checklist including the location of each item in the manuscript. -The purpose of the review, including how the collected data will be used, should be clearly addressed in the abstract and the introduction. -The authors should indicate where they are in the process of conducting the review and provide dates (either past if started or anticipated if not). METHODS AND ANALYSIS  -Aim: If the aim of the review is to identify interventions associated with improved outcomes specifically, the authors should define what constitutes an “improved” outcome on the outcome measures. Search strategy:  -It would be helpful to provide additional detail with respect to how the search strategy was developed (e.g., which authors were involved, any involvement of a research librarian, any specificity/sensitivity testing performed of search terms during search development). -Ideally the database search strategy would be reviewed by e.g., a research librarian, and PRESS is an option for this. If this is not done, it should be noted as a limitation. Grey literature:  -It is unclear how and why the grey literature sources were selected, and how the results will be used in this review (e.g., newspaper articles?). Outcome measures:  -If the primary outcome is a composite outcome, this should be explicitly stated. -The outcomes are clearly specified, but it is unclear how the outcome data collected is going to be synthesized and used, particularly if there is insufficient data to conduct a meta-analysis, which the authors raise as a likely possibility.
--

	Study screening -It is standard for titles to be screened in addition to abstracts. Will titles be screened? -How will inter-rater reliability be assessed between reviewers, and how will this be handled if it is low? Data collection -It is unclear if data extraction will be split between two reviewers (if so, what if any quality control measures will be in place?) or if performed by two reviewers in duplicate (if so, how will any discrepancies be resolved?)
--	--

VERSION 1 – AUTHOR RESPONSE

Reviewer 1

1 Background; This presents a good overview of the problem of comparatively high rates of death in LMIC compared to HIC for gastroschisis. Could authors include a paragraph or so more clearly linking this burden up with the justification and need for the review? This [link] doesn't seem to come out very clearly.

Response:

Further information has been added to the introduction section to help clarify this. Introduction

2 In page 4 line 21, could authors include 'Protocols' so that the statement reads "Preferred Reporting Items for Systematic review and Meta-Analysis Protocols"(PRISMA-P)

Response: This has been revised accordingly.

3 In the data synthesis, given the anticipated variation in interventions authors are likely to meet, you mention descriptive statistics would be reported. This should be accompanied with a narrative synthesis.

Response: This has been added.

4 Secondly, authors mention a meta-analysis is not planned, but in case there is 'appropriate data' they will progress with a meta-analysis. Could you be more specific and or elaborate on what 'appropriate data' you will be interested in so as to perform a meta-analysis?

Response: Appropriate data will be defined as: two or more studies comparing the mortality between two or more of the same interventions so we can pool the data and perform a meta-analysis. For example two or more studies comparing the mortality outcome between intervention 'a' with intervention 'b'. Details have been added to the protocol accordingly.

5 Consider discussing your process of data management.

Response: Details added. Study screening, data extraction and data synthesis

Reviewer 2

No changes required

Reviewer 3

1 The authors indicate the protocol was prepared using PRISMA-P checklist. The authors should provide the completed PRISMA-P checklist including the location of each item in the manuscript.

Response: This has been done .

2 The purpose of the review, including how the collected data will be used, should be clearly addressed in the abstract and the introduction.

Response: Edits have been made to both the abstract and introduction.

3 The authors should indicate where they are in the process of conducting the review and provide dates (either past if started or anticipated if not).

Response: This has been added to the title page beneath the registration details. 4

Aim: If the aim of the review is to identify interventions associated with improved outcomes specifically, the authors should define what constitutes an “improved” outcome on the outcome measures.

Response: Definitions have been added.

5 -It would be helpful to provide additional detail with respect to how the search strategy was developed (e.g., which authors were involved, any involvement of a research librarian, any specificity/sensitivity testing performed of search terms during search development).

-Ideally the database search strategy would be reviewed by e.g., a research librarian, and PRESS is an option for this. If this is not done, it should be noted as a limitation.

Response: Further details regarding the search strategy have been added. A research librarian was involved.

Details regarding which authors were involved have been added to the contributions section.

Sensitivity and specificity testing was undertaken by NW and EW and we did change the original search strings based on this. Initially we had 1) the conditions, 2) the context – LMICs, 3) the interventions. In order to improve specificity and sensitivity we changed the search to 1) the population 2) conditions 3) context. This made it more specific to neonates, which is the population we are interested in and more sensitive to a broader range of interventions since all could be included, not just those listed in the original search string 3.

6 -It is unclear how and why the grey literature sources were selected, and how the results will be used in this review (e.g., newspaper articles?).

Response: This has been addressed in the text now.

7 Outcome measures:

-If the primary outcome is a composite outcome, this should be explicitly stated.

Response: The primary outcome is not a composite outcome. The primary outcome is mortality. Since mortality indicators vary between studies we shall include the three most commonly used indicators. If a meta-analysis is to be undertaken we will either compare studies utilising the same mortality indicators or we shall adjust accordingly with expert input.

8 -The outcomes are clearly specified, but it is unclear how the outcome data collected is going to be synthesized and used, particularly if there is insufficient data to conduct a meta-analysis, which the authors raise as a likely possibility.

Response: Details have been added to the 'Data Synthesis' section. See also response to Reviewer

9 -It is standard for titles to be screened in addition to abstracts. Will titles be screened?

Response: Titles were also screened. A sentence has been added stating this.

10 -How will inter-rater reliability be assessed between reviewers, and how will this be handled if it is low?

Response: Details have been added. Study screening and data extraction

11 -It is unclear if data extraction will be split between two reviewers (if so, what if any quality control measures will be in place?) or if performed by two reviewers in duplicate (if so, how will any discrepancies be resolved?)

Response: Details have been added to the Data Extraction section.